# Viscoelastic Testing and Coagulopathy of Traumatic Brain Injury

**DOI:** 10.3390/jcm10215039

**Published:** 2021-10-28

**Authors:** Jamie L. Bradbury, Scott G. Thomas, Nikki R. Sorg, Nicolas Mjaess, Margaret R. Berquist, Toby J. Brenner, Jack H. Langford, Mathew K. Marsee, Ashton N. Moody, Connor M. Bunch, Sandeep R. Sing, Mahmoud D. Al-Fadhl, Qussai Salamah, Tarek Saleh, Neal B. Patel, Kashif A. Shaikh, Stephen M. Smith, Walter S. Langheinrich, Daniel H. Fulkerson, Sherry Sixta

**Affiliations:** 1Department of Neurosurgery, Indiana University School of Medicine, Indianapolis, IN 46202, USA; jlbradbu@iupui.edu; 2Department of Trauma Surgery, Memorial Hospital, South Bend, IN 46601, USA; SThomas@beaconhealthsystem.org; 3Department of Emergency Medicine, Indiana University School of Medicine—South Bend, South Bend, IN 46617, USA; nikrsorg@iu.edu (N.R.S.); ashmoody@iu.edu (A.N.M.); singh64@iu.edu (S.R.S.); 4Department of Intensive Care Medicine, St. Joseph Regional Medical Center, Mishawaka, IN 46545, USA; nmjaess@nd.edu (N.M.); mberquis@nd.edu (M.R.B.); toby.brenner@myemail.indwes.edu (T.J.B.); jlangford@butler.edu (J.H.L.); malfadhl@iu.edu (M.D.A.-F.); kacysalama@yahoo.com (Q.S.); Drsaleh20@gmail.com (T.S.); 5Department of Otolaryngology, Portsmouth Naval Medical Center, Portsmouth, VA 23708, USA; mkmarsee@gmail.com; 6Department of Neurosurgery, Memorial Hospital, South Bend, IN 46601, USA; npatel@beaconhealthsystem.org (N.B.P.); kshaikh@beaconhealthsystem.org (K.A.S.); SSMITH5@beaconhealthsystem.org (S.M.S.); WLangheinrich@beaconhealthsystem.org (W.S.L.); dhfulkerson@beaconhealthsystem.org (D.H.F.); 7Department of Neurosurgery, St. Joseph Regional Medical Center, Mishawaka, IN 46545, USA; 8Department of Trauma Surgery, Envision Physician Services, Plano, TX 75093, USA; sherry.sixta@envisionhealth.com

**Keywords:** adenosine diphosphate, arachidonic acid, blood platelets, brain injuries, traumatic, cerebral hemorrhage, critical care, fibrinolysis, mortality, resuscitation, thromboelastography

## Abstract

A unique coagulopathy often manifests following traumatic brain injury, leading the clinician down a difficult decision path on appropriate prophylaxis and therapy. Conventional coagulation assays—such as prothrombin time, partial thromboplastin time, and international normalized ratio—have historically been utilized to assess hemostasis and guide treatment following traumatic brain injury. However, these plasma-based assays alone often lack the sensitivity to diagnose and adequately treat coagulopathy associated with traumatic brain injury. Here, we review the whole blood coagulation assays termed viscoelastic tests and their use in traumatic brain injury. Modified viscoelastic tests with platelet function assays have helped elucidate the underlying pathophysiology and guide clinical decisions in a goal-directed fashion. Platelet dysfunction appears to underlie most coagulopathies in this patient population, particularly at the adenosine diphosphate and/or arachidonic acid receptors. Future research will focus not only on the utility of viscoelastic tests in diagnosing coagulopathy in traumatic brain injury, but also on better defining the use of these tests as evidence-based and/or precision-based tools to improve patient outcomes.

## 1. Introduction

### 1.1. Incidence of Coagulopathy of Traumatic Brain Injury

Occult coagulopathy of traumatic brain injury (TBI) reportedly affects a high percent of trauma patients with a significant increase in morbidity [1,2,3,4,5,6]. Literature frequently cites an estimate that “one-third” of patients with a TBI will develop a coagulopathy of TBI (CTBI). However, the true incidence of coagulopathy reported in these patients has been cited as anywhere from 7 to 63%. This variability arises from the lack of consistency in the definition of coagulopathy and its causes in TBI. Thus, comparisons of diagnosis and treatment among different populations of CTBI patients remain problematic [6,7,8,9,10,11,12,13,14,15,16,17,18,19,20,21,22,23,24,25,26,27,28,29,30,31,32,33,34,35,36,37,38,39,40,41].

### 1.2. Implications of CTBI and Relation to VET-Based Definition

In addition to its fluctuant frequency, CTBI is defined by variable cut-off values when using common coagulation assays (CCAs), such as platelet count, prothrombin time (PT), international normalized ratio (INR), partial thromboplastin time (PTT), and fibrinogen levels. CCAs are also limited to detecting the initiation of clot formation and fail to provide information regarding the strength and integrity of the clot formed [4,6,23,42,43,44,45,46,47,48]. Furthermore, CCAs are not sensitive detectors of hemostatic integrity in patients with multiple systemic polytrauma, including TBI, and fail to predict coagulopathy in patients on pre-injury anticoagulant medications, particularly antiplatelet drugs. Prescriptions for anticoagulants are only continuing to rise due to the increased prevalence of cardiovascular disease and an overall aging population [23,49]. Moreover, patients experience up to a 30-fold increased risk of disability and morbidity when compared to TBI patients without the development of coagulopathy [31,50,51,52,53,54,55,56,57]. These facts emphasize the need for a quick and accurate test to identify an abnormal coagulation profile and provide rapid management of coagulopathy if indicated [40].

On the other hand, whole blood hemostatic assays known as viscoelastic tests (VET)s give a more detailed and rapid view of hemostatic integrity by providing a point of care (POC) view of the lifespan of a clot. Published in 2019, the 5th edition of the European guideline on management of major bleeding and coagulopathy following trauma changed their recommendations to include the use of VETs such as thromboelastography (TEG^®^) and rotational thromboelastometry (ROTEM^®^) for patients with systematic multiple trauma and with TBI [47]. There are many advantages in using VETs to detect CTBI [4,6,23,48,58]. For example, VETs provide real-time coagulation information on the presence of anticoagulation or antiplatelet medications and the patient’s initial coagulation profile, allowing for the monitoring of therapeutic interventions such as hemostatic adjuncts or blood component transfusion [23,47,58,59,60,61,62,63]. Furthermore, modified TEG^®^ Platelet Mapping (TEG-PM^®^) and ROTEM^®^ with adjunctive multiple electrode aggregometry (MEA) can also be used to quickly determine platelet function abnormalities from inherent TBI coagulopathy or antiplatelet medications [23,39,58,61,62,63,64,65,66,67].

### 1.3. Inadequacy of Conventional Coagulation Assays in the Diagnosis of CTBI

CCAs have historically been the cornerstone of CTBI diagnosis, and this holds true even as recently as a review from late 2020, citing the use of CCAs because of a paucity in literature regarding VET use in CTBI [50]. Yet, CCAs lack of sensitivity in determining hemostatic derangement may contribute to under-diagnosis of coagulopathy in trauma patients [41,68].

Because of their “global” nature, VETs such as ROTEM^®^, TEG^®^, and thrombin generation tests may provide more detailed and useful data concerning the overall ability to achieve hemostasis [6]. These global hemostatic assays use a surrogate endpoint of maximum clot firmness and allow the evaluation of additional information on clotting kinetics, platelet-fibrin interactions, and fibrinolysis [69]. In addition, POC non-VET platelet function tests like Platelet Function Analyzer-100 (PFA-100), MEA (also known as Multiplate^®^), VerifyNow P2Y12, and the modified VET TEG-PM^®^ may assist in detecting platelet dysfunction [39,58,63,64,67,70,71,72,73,74]. The degree of ADP-receptor inhibition has been suggested to correlate with the severity of TBI, described as a “dose-response relationship” between TBI severity and the degree of platelet dysfunction [64]. There may also be a role for platelet function assays in monitoring platelet dysfunction mediated by antiplatelet agents [23,52,58,61,64,66,67,74].

In both isolated TBI and polytrauma TBI patients, VET assays provide real-time assessment of hemostasis and prognostication [6,44,45,46,47,60,75,76]. Compared to CCAs, the rapidity at which “global” hemostatic assays predict outcome and severity in a population of patients with severe TBI is significant [6,77]. Because of these more timely and accurate results, VETs offer additional opportunities to correct hemo-coagulative defects and, ultimately, improve patient outcomes. With these capabilities, VETs like ROTEM^®^ and TEG^®^ are gaining acceptance in clinical practice and represent a mechanism to improve patient care [23,47,78,79,80].

VET analysis of the coagulopathic spectrum of CTBI reflects the pathophysiology of that spectrum whether associated with isolated TBI or with TBI complicated by trauma-induced coagulopathy [23,47,48,60,61,62,81].

## 2. Pathophysiology in CTBI and Its Relation to VETs

CTBI is associated with a disorder of primary hemostasis, requiring early utilization of platelet function testing to define the coagulopathy. The progression of platelet dysfunction can be monitored by VETs [23,48,53,56,60,82].

After the initial TBI, platelet receptors demonstrate early inhibition, and the degree to which this occurs is a function of the TBI severity; mortality is irrespective of total platelet count. This inhibited platelet phenotype is indicated by a malfunction in platelet receptors for adenosine diphosphate (ADP) or arachidonic acid (AA) [1,53,83,84,85,86] as well as other platelet receptors such as collagen, ristocetin, Thrombin receptor activating peptides (TRAP), and protease activated receptor-1 (PAR-1) in TBI [39,65,87].

Xu et al. suggested TBI elicits two distinct processes that may cause coagulopathy: local primary hemostasis to the injured site and systemic endotheliopathy in part attributable to relatively high levels of von Willebrand factor (vWF) stored in brain tissue [88]. First, the vWFs are immobilized at the injured site causing rapid local platelet activation to seal off the vascular injury. Second, the damage in the blood-brain barrier (BBB) causes injured brain cells to release extracellular vesicles comprising procoagulant molecules, notably brain tissue factor (TF) (which is usually isolated from blood), which then further activates the remote endothelial cells and platelets systemically to release more stored uncleaved vWF and exacerbating systemic pro-coagulation [88,89].

The endothelial dysfunction of CTBI then results in a crosstalk between dysfunctional platelets and endothelium. This is the earliest manifestation of CTBI, first demonstrated by VETs, which quantify dysfunction of varying platelet receptors. This CTBI pathophysiology often results in hemorrhagic expansion. Therefore, the pathophysiology of CTBI can be more effectively assayed by VETs with specialized function analysis than by CCAs, as has been shown for multiple systemic trauma [23,29,38,68,83,84,90,91,92,93,94,95,96,97,98]. Because of this early platelet hyperactivation and subsequent hypercoagulation, the consumption and/or exhaustion of platelets and coagulation factors leads to a decrease in fibrinogen, often resulting in the later onset of bleeding [23,52,53,99,100].

In addition, a significant number of patients with CTBI have suppressed fibrinolysis, termed ‘fibrinolytic shutdown’. True hyperfibrinolysis as manifested by VETs is uncommon and has been supported by the recent failure of studies to confirm the findings of the CRASH-3 trial, demonstrating a small therapeutic benefit of the antifibrinolytic tranexamic acid. It has been proposed that VETs are not sensitive enough to determine occult fibrinolysis. There may be intracerebral fibrinolysis not detected by in vitro measurement of whole blood VET samples [95,101,102,103,104].

## 3. Basics of TEG^®^/ROTEM^®^

### 3.1. Description of the Cup and Pin

TEG^®^ and ROTEM^®^ tracings depict hemostatic integrity, measuring initiation, amplification, propagation, and termination of whole blood to form a clot. Figure 1 demonstrates and describes the pin and cup procedure for carrying out TEG^®^ and ROTEM^®^ testing [43,105,106,107,108]. TEG^®^ and ROTEM^®^ devices apply a rotation and measure the change in tension on the pin within the cup as the clot forms (Figure 1). This is plotted over time, generating the parameters of TEG^®^ and ROTEM^®^ (Figure 2) [43,106,107].

### 3.2. TEG-PM^®^ and ROTEM^®^ with Specialized Platelet Function Testing to Diagnose and Guide Platelet Transfusion in Patients with CTBI

Trauma can induce platelet dysfunction, which may not be detectable by standard ROTEM^®^ or TEG^®^ assays. Platelet function alone can be studied using TEG-PM^®^ using heparin, factor XIII, reptilase, AA, and ADP to form platelet-fibrin clots independent of thrombin as demonstrated in Figure 3 [109]. Figure 4 shows the platelet mapping tracing superimposed on a physiologic TEG^®^ tracing.

In trauma patients and TBI patients, TEG-PM^®^ AA inhibition is significantly higher than in healthy controls [1,110]. In TBI versus non-TBI trauma patients, TEG-PM^®^ AA inhibition is also significantly higher [110]. CCAs and standard TEG^®^ variables (R, α, MA) are relatively normal, while TEG-PM^®^ ADP inhibition is significantly higher in patients with isolated TBI versus control subjects. Platelet dysfunction is also significantly higher in patients with severe versus mild-to-moderate TBI [1,53,63,64].

ROTEM^®^ also has its own platelet functionality testing called PLTEM, which has not been studied in TBI and is rarely used in the setting of polytrauma [113,114,115]. In PLTEM, the A10 parameter represents the amplitude of the tracing 10 min after the end of CT. PLTEM calculates the difference between the EXTEM A10 (A10_EX_) and the FIBTEM A10 (A10_FIB_). The A10_EX_ represents the clot strength of the entire extrinsic pathway and the A10_FIB_ represents the same pathway but without the contribution by platelets. Thus, the PLTEM represents the contribution of platelets to clot strength. Reporting the PLTEM parameter would likely involve manually calculating and extracting A10 data from each assay [115,116]. For those situations where TEG-PM^®^ and PLTEM are not available, ROTEM^®^ has adopted an accompanied MEA platform, also known as MultiPlate^®^, for determining activity of platelet function in TBI. The MultiPlate^®^ analyser—after the platelets are activated by a specific agonist in each well, such as ADP, AA, ristocetin, collagen, or thrombin receptor activating peptide (TRAP)—operates by the platelets adhering to the electrodes and reducing impedance (Figure 5). Platelet aggregation and activation are described as an area under the curve in relation to a standard control baseline. This MEA test is not a viscoelastic test but is used as an adjunct with ROTEM^®^ to isolate platelet dysfunction [39,65,66,87,117,118,119]. Intradevice variability is low for TEG-PM^®^ and MultiPlate, but the measurement of platelet function overall correlates poorly in injured trauma patients [120].

MEA is affected by platelet transfusion and COX inhibitor treatment. In TBI, MEA has demonstrated platelet dysfunction in patients with and without COX inhibition treatment. For those on platelet inhibitors—a common subset of TBI patients—MEA generally shows low values initially and increases within 48–72 h [39,122,123]. However, compared with antiplatelet agents, later studies of platelet transfusions showed no significant difference in outcome [124,125]. Hence, the substantial heterogeneity in results regarding the efficacy of platelet transfusion for TBI with and without preinjury antiplatelet therapy remains an area of fertile research and controversy [126].

In addition, other non-viscoelastic assays have been added as adjuncts to assist in the determination of the adequacy of platelet function for these patients. These electrochemical and biochemical tests, including PFA-100 and VerifyNow Aspirin/P2Y12 assays, have been adapted from the cardiology and neurological population of patients whose blood required determination of platelet inhibition following administration of antiplatelet agents. These tests have demonstrated heterogeneous results when used to gauge platelet dysfunction following TBI for patients with or without preinjury antiplatelet medications [2,63,72,74,120]. Notably, these assays have several shortcomings, including limited availability and methods that have not been subject to large-scale quality control. The tests themselves are of little value in cases of low platelet counts. The assay-based monitoring of direct oral anticoagulant (DOAC) effectiveness is also in its infancy. The utilization and application of whole blood POC assays are still not universally practiced in trauma [127].

## 4. Utilizing VETs for the Diagnosis and Treatment of TBI

There are three main areas in which TEG^®^ has been used to analyze TBI, representing the stages of clinical evaluation assisted by VETs: diagnosis of CTBI, management of CTBI, and prognosis of TBI [4,23,129,130,131,132].

### 4.1. VETs to Diagnose, Treat, and Prognosticate CTBI

#### 4.1.1. Diagnosis of CTBI

TEG^®^ and TEG-PM^®^ reportedly assist in diagnosis and differentiation between coagulopathies of TBI and non-TBI patients. TEG^®^ and ROTEM^®^ parameter have been found to add sensitivity to the diagnosis of CTBI with abnormalities of all parameters reported [5,37,53,60,86,100,110,133,134,135,136,137,138,139].

#### 4.1.2. Treatment of CTBI

TEG^®^ and TEG-PM^®^ can be used to guide the diagnosis and treatment of patients with CTBI [53,61,62,64,129,140,141,142,143]. These TEG-guided therapies enable physicians to quickly deliver focused therapies, accurately correcting coagulopathy while potentially conserving blood products. Still, there are heterogeneous and conflicting results regarding whether application of VETs like TEG^®^ correlate with improved patient outcomes and reduced mortality [1,39,40,84,86,123,129,140,141,144,145,146]. Several studies exhibit improved survival when using a TEG-based resuscitation strategy compared to a CCA-guided treatment (PT, aPTT, fibrinogen, and D-dimer) [56,139,147].

For example, a TEG-based resuscitation strategy was shown to improve survival in a pragmatic randomized controlled trial as compared to one based on CCAs; however, subgroup analysis on TBI patients did not demonstrate reduced mortality. It should be noted that the TBI subgroup was not specified a priori nor sufficiently powered for this outcome [129]. In addition, the iTACTIC study indicated possible improvement and survival in the subgroup of injured multiple trauma patients with TBI whose resuscitation was guided by ROTEM^®^ with specific emphasis on FIBTEM-guided resuscitation using soluble fibrinogen concentrate [140]. Improved clot quality with decreased neurosurgical reintervention and decreased incidence of progressive hemorrhagic injury PHI has been noted with ROTEM^®^ [45]. Delayed clot formation without associated fibrinolytic abnormalities is a singular manifestation of the unique hemostatic derangement of severe isolated TBI. Activated coagulation time (ACT) when prolonged, and coupled with low fibrinogen levels, suggest that early coagulation factor replacement may be more critical than empiric antifibrinolytic therapy. It is clear the mechanisms that precipitate coagulopathy in TBI differ from those mechanisms in multisystem trauma and warrant further investigation [60].

**Figure 5 jcm-10-05039-f005:**
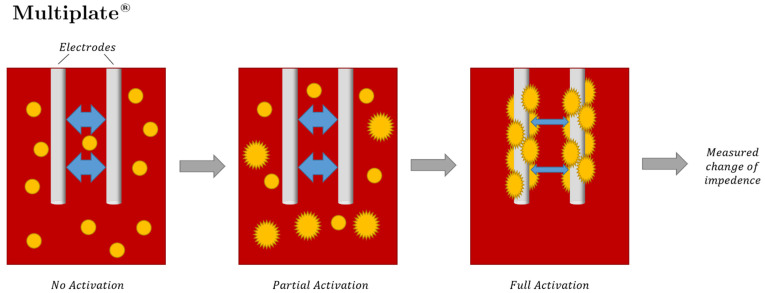
The Multiplate^®^ multiple electrode impedance aggregometer measures platelet function in diluted anticoagulated whole blood. Each plate uses two sensors containing a pair of silver-coated copper wire electrodes. Over a set duration of time, aggregation of activated platelets on the surface of the electrodes causes a measurable change in impedance, measured in aggregation units (AU, where 8AU ≈ 1 Ω) and area under the AU curve [148].

It has also been suggested that POC testing with platelet function analysis be used to manage TBI patients who need extracorporeal membrane oxygenation [149].

#### 4.1.3. Prediction of Morbidity and Mortality in CTBI

TEG^®^ and TEG-PM^®^ assays can predict a range of clinical outcomes in patients with CTBI, including bleeding complications, mortality, and neurosurgical intervention [1,3,56,72,84,86,139,144,145,150,151,152,153]. However, Rapid TEG^®^ parameters have been associated with the increased odds of hematoma expansion in patients with TBI and SAH [153].

Several studies have demonstrated that TBI patients are significantly more hypercoagulable at presentation, likely due to increased platelet aggregation. Moreover, initial hypercoaguability determined by TEG^®^ in TBI patients was associated with prolonged ICU stays, longer overall hospital admissions, and increased mortality [3,5,53,56,64,110,135,136,137]. TBI patients with initial hypocoagulopathic TEG^®^ values (longer reaction times, flat alpha angles, thin MAs) highly correlate with mortality in isolated TBI [86].

Standard TEG^®^ assays conducted at admission and after 6 h of hospitalization on TBI patients have been able to identify patients who will experience worse clinical outcomes. TEG^®^ monitors the rate of clot breakdown in addition to clot formation: an increased rate of fibrinolysis and abnormal clot formation identifies those patients who may require a neurosurgical procedure or have a greater risk of worse mortality [152]. In addition, prognosis and progression of intracranial bleeding are dependent on coagulopathy identified by admission TEG^®^ parameters but not CCAs [3,40,82,86,139,154].

### 4.2. Basic TEG^®^/ROTEM^®^ Parameters Triggering BCT and HAT

Prolongation of reaction time (R) in TEG^®^ or clotting time (CT) in ROTEM^®^ indicates a coagulation factor deficiency and warrants fresh frozen plasma (FFP) and/or prothrombin complex concentrate administration [47,155]. A prolonged clot formation/kinetics (K) in TEG^®^ or clot formation time (CFT) in ROTEM^®^ and a low ɑ-angle reflect a fibrinogen or fibrin production deficiency, warrants cryoprecipitate or fibrinogen concentrate administration. A low maximum amplitude (MA) in TEG^®^ or maximum clot formation (MCF) in ROTEM^®^ depicts decreased clot formation and warrants platelet repletion. An increase in lysis at 30 min (LY30) in TEG^®^ or a decrease in lysis index at 30 min (LI30) suggests fibrinolysis and is treated with an antifibrinolytic agent [43,44,105,106,107,155,156,157,158]. Currently, thresholds for platelet administration to patients with TBI are in a state of evolution [23,40,52,61,63,64].

### 4.3. Guiding Blood Products with VETs in TBI

Recommended thresholds have been proposed regarding VETs for guiding the administration of blood components and hemostatic therapy for hemorrhaging trauma patients without TBI [46,47,63,80,159,160]. A preliminary study by Furay et al. suggested improved survival in TBI patients with VET-guided BCT resuscitation [61]. For example, as mentioned above, the 2021 ITACTIC study suggests that survival of TBI patients may be improved with the use of VET-guided treatment algorithms [140]. For patients with TBI, whether isolated or associated with multiple trauma, increased survival rates could be a product of individually-tailored and prompt administration of blood products, which, as we have demonstrated, can be appropriately guided by VETs [129]. An analysis in the Prehospital Air Medical Plasma (PAMPer) trial upheld this conclusion, finding that a subgroup of TBI patients exhibited increased survival with the administration of pre-hospital plasma guided by the TEG^®^ [161,162]. Reductions in cerebral inflammation, intracerebral bleeding, and cerebral ischemia are attained using a targeted management approach, thus potentially contributing to the documented improvement in survival [140].

#### 4.3.1. VET-Guided FFP, Fibrinogen Concentrate, and Cryoprecipitate in TBI

The use of FFP in patients with moderate and severe TBI is under investigation [163,164,165]. Both empirical infusion of FFP in severe-TBI patients and use of FFP in patients with TBI and moderate coagulopathy (with or without pRBCs) have been associated with poor functional outcomes [125]. However, there are survival benefits with early FFP administration in patients with multifocal intracerebral hemorrhage (ICH) [163] or with ratio-based blood-product transfusion in patients suffering from TBI when guided by the TEG^®^ and ROTEM^®^ [129,140,165].

A plasma-based coagulation resuscitation strategy also suggests that such administration should be avoided in patients without substantial bleeding [46]. Instead, in such cases, it has been found that therapy guided by TEG^®^ using PCC and fibrinogen concentrate for treatment is more advantageous than FFP use [6,23,44,140,166]. Fibrinogen concentrates are efficient and should be used primarily for those patients with bleeding complications. If unavailable, FFP or cryoprecipitate may be used instead. Replenishment of fibrinogen can be guided by ROTEM^®^ MCF values [167].

Fibrinogen is defined as the substrate for clot formation, playing a key role in hemostasis through conversion to fibrin, undergoes crosslinking, and interacts with platelets [168]. TBI causes coagulation factor consumption to significantly increase; in tandem, this causes consumption of fibrinogen. Fibrinogen concentrations thereafter do not recover beyond normal amounts until at least 2–3 days after the initial TBI [169]. Significantly reduced fibrinogen concentrations in the initial phase following an injury not linked to TBI have also been observed to be associated with high rates of mortality [170]. Due to this linkage, concentrations should remain within 1.5–2 g/L through the utilization of either cryoprecipitate or fibrinogen concentrates [46]. Late-stage TBI can also cause plasma concentrations of fibrinogen to increase. This increase in fibrinogen—which is an acute phase reactant and a marker of post-resuscitative inflammation—is a late compensatory reaction to the previously hypocoagulopathic state for patients with severe TBI, which is similar to the increase in fibrinogen found in any patient in the period of recovery following successful resuscitation after hemorrhage [171]. This increase in plasma fibrinogen as a marker of inflammation correlates with an increase in cerebrovascular permeability and a resultant injury to the penumbra that ultimately serves as a gateway to further inflammation [171].

As a result of the above-mentioned studies, the most recent European guidelines recommend utilization of VETs to guide FFP, fibrinogen concentrate, and cryoprecipitate for TBI [47].

#### 4.3.2. VET-Guided Diagnosis and Treatment of Platelet Dysfunction in TBI

Platelets’ function consists of adhesion, activation, and aggregation which initiate the action of coagulation factors to reach hemostasis. TF initiates coagulation and platelets contribute to enhance the reaction by providing a negatively charged surface which allows adhesion, activation, and aggregation. Their enhancement of coagulation with tissue damage and TF release as a trigger, therefore plays an important role in the pathophysiology of TBI.

TBI affects both platelet number and function [40]. Low platelet count (<100,000/mm^3^) has been shown to be an independent predictor of mortality in TBI patients [172]. However, as has been described above in the section on pathophysiology, platelet dysfunction at the ADP and AA receptors (as measured by the TEG-PM^®^) reflects the severity of TBI even with patients with normal platelet counts. In addition, ROTEM^®^ using the MEA and the VerifyNow P2Y12 to quantify platelet dysfunction in TBI has demonstrated similar dysfunction at the AA and ADP receptors as well as the ristocetin, PAR1, and collagen platelet receptors in patients with TBI [1,53,56,60,65,83,84,85,86,87]. Furthermore, the TEG-PM^®^ showed elevated ADP inhibition, which is further linked with increased mortality, when measured directly after an isolated blunt TBI [144]. The TEG-PM^®^ was also successful at reporting elevated ADP inhibition with increasing injury severity. This correlation between the severity of injury and the level of inhibition of the predominate ADP receptor has been defined as the “dose-response curve” describing the relationship between the level of injury dose and the response to that injury as determined by ADP inhibition response [52,64]. As a result, it has been noted that TBI patients with severe (greater than 60%) ADP inhibition had increased mortality compared to other TBI patients with less ADP inhibition [151]. In parallel, TBI patients who suffered from a bleeding event had a higher AA inhibition compared to other TBI patients [84]. A weak correlation was established between TEG-PM’s AA inhibition measure and ICH progression when there was no significant correlation for other platelet assays such as Multiplate aggregometry and VerifyNow™ [72].

It is still unclear whether the presence of ADP inhibition in this patient population is a prognostic indicator or a therapeutic target [61].

Trauma-induced platelet dysfunction can be corrected using TEG-PM^®^ to guide the treatment plan. A TEG-PM^®^ based algorithm has been found to significantly decrease the number of platelet transfusions needed for older TBI patients who are on anti-platelet medications [4,82]. Patients with platelet dysfunction on admission suffering from severe TBI have also been studied [1,53,61,83,84,86,87]. Using a protocol, platelet dysfunction was demonstrated by an ADP-inhibition of greater than 60%. These patients received one unit of apheresis platelets and exhibited decreased mortality compared to patients who were not given these platelets. Furthermore, the use of TEG-PM^®^ was able to limit the total overall administration of blood products efficiently [61,86]. TBI-associated platelet transfusion remains controversial. Moderate thrombocytopenia outcomes were not improved by post TBI platelet concentrate transfusions [125]. Additionally, platelet transfusions performed on patients undergoing antiplatelet therapy at the time of a mild TBI occurrence failed to improve outcomes, but evidence may link this with thrombotic complications. Currently, retrospective registry studies have failed to show that patients undergoing antiplatelet preinjury would benefit from routine platelet transfusions after sustaining traumatic ICH [49,124].

Despite significant literature that demonstrates platelet dysfunction associated with TBI for patients with and without preinjury and antiplatelet use, there is little consensus regarding the indication for the administration of platelets for those patients. Observational and retrospective studies have shown some benefit when platelet transfusion is guided by TEG-PM^®^ and ROTEM^®^ with MEA, PFA, or VerifyNow P2Y12. This remains an area of significant research since the administration of platelets in these patients is now determined locally by institutional preferences with variable use of TEG-PM^®^ and ROTEM^®^ with adjunctive PFAs [23,39,58,74,173,174,175].

In addition, the increased use of anticoagulants and antiplatelet agents in patients with TBI has led to widespread and institution-dependent platelet transfusion with such injuries [23,40,52,61,63,64]. There are many side effects that can occur with platelet transfusion which makes this practice controversial, such as sepsis, transfusion-related acute lung injury, blood group type incompatibility reactions, arrhythmia, stroke, and death [176,177]. Hence, in patients who suffer from TBI and are taking antiplatelet medication, the advantages and disadvantages of platelet transfusion need to be carefully considered before platelet transfusions are administered.

A plethora of assays has demonstrated that aspirin causes a decrease in platelet aggregation. The Aspirin Response Test is used to show the impact of aspirin on platelet function and has determined that 64% of TBI patients on aspirin who have been administered platelets have a reversal of platelet inhibition [74]. However, the adverse effects of transfusing platelets may outweigh the benefits of the physiological response. When assessing mortality in patients with TBI who received a platelet transfusion versus those who did not receive a platelet transfusion, there was no significant difference in mortality between the two groups. In addition, increased mortality was observed when patients on pre-injury antiplatelets were given platelets for TBI related coagulopathy. As severity of injury increased, the transfusion rate of platelets increased as well [178]. No significance was found between platelet transfusion and need for surgical intervention, rate of neurologic decline, progression of injury based on GCS, cardiac events, respiratory events or imaging [179]. A multi-institutional observational study found that aspirin-related platelet inhibition was significantly decreased with platelet transfusion. However, platelet transfusions did not impact mortality of these TBI patients [142]. Platelet transfusion may seem to be an intuitive therapy for platelet dysfunction, but for TBI patients on antiplatelet agents, it is not effective at improving outcomes [146]. The large PATCH trial recently has demonstrated increased mortality in patients on antiplatelet agents who had TBI. This study hypothesized that whatever benefit accrued from the administration of platelets in patients on antiplatelet agents with TBI were counteracted by the effect of a local hypercoagulable state at the microvasculature in the penumbra of injured brain tissue. For this reason, and for the smaller studies that demonstrate the benefit of VET guided platelet transfusion in patients with TBI and platelet dysfunction, there is increased interest in not only the use of VET to guide platelets in patients with TBI and preinjury antiplatelet agents, but also in using desmopressin to enhance platelet function in these patients [23,52,58,61,126,144,173,174,175,180]. DDAVP has been shown to stabilize platelet dysfunction in neurosurgical patients and those with spontaneous ICH with abnormal platelet activity who were previously on aspirin. Because of the heterogeneous reports regarding the incidence of platelet dysfunction in isolated TBI for patients with and without prior antiplatelet agents, further research is clearly indicated regarding the incidence of platelet dysfunction in this population and whether replenishment of platelets and/or desmopressin is of therapeutic benefit [62,82,173,181,182].

Other studies have compared the use of desmopressin to platelet transfusion in patients with severe TBI. Both treatments were found to improve ADP inhibition similarly while displaying no differences in mortality. However, treatment with platelets did exhibit TEG^®^ parameters (α, G, MA) that were corrected to a greater degree, and a greater increase in clot strength [62,83]. Platelet transfusion has also been shown to improve AA inhibition specifically in patients with blunt TBI who were taking antiplatelet agents prior to injury. However, no improvement in mortality was seen with this treatment [142,152].

The above-mentioned studies concerning the use of VETs with and without adjunctive platelet function tests demonstrate heterogeneous results, which has resulted in institutional preferences for guiding platelet transfusion for patients with TBI (Table 1).

### 4.4. Preinjury Antithrombotic Use

Since the TEG^®^ and ROTEM^®^ have been mostly used for patients with multiple trauma who need immediate BCT and HAT, there has been sparse literature for VET in TBI for a small group of patients who are on anticoagulants such as warfarin and DOACs. While it is well known that TEG^®^ and ROTEM^®^ do not measure warfarin or DOACs hemostatic activity, the existing literature is limited to a few studies of modified VETs that allow for analysis but are not used clinically.

Increased incidence of patients with comorbidity is observed as the TBI demographic switches to an older age [122,183]. Modern treatment of coronary artery and chronic cerebrovascular disease indicates a need for these patients to take antiplatelet or anticoagulant drugs; both of which are associated with increased bleeding and, ultimately, worsened TBI outcomes [122,184,185,186,187,188,189]. Meta-analysis on 49 patients supplementing warfarin at the occurrence of TBI shows that the risk of poor outcome is double that of those not supplementing warfarin. However, similar demographic analysis on patients supplementing antiplatelet therapy did not indicate a clear increase in risk to those not undergoing antiplatelet therapy [184,188,189]. Additional retrospective evidence reiterates this observation [186], yet other studies argue that antiplatelet therapy supplementation preinjury, especially in an older demographic, could result in nearly a twofold increase in the occurrence of traumatic ICH even after minor TBI, when compared with patients not supplementing antiplatelet therapy at the time of the injury [185,190,191]. Preinjury warfarin or clopidogrel are independent factors from the severity of TBI for prediction of disease progression, ICH, and worse prognosis [36,124,185,192,193]. Currently, the risks that patients supplementing newer, target-specific DOACs endure in the face of TBI is unknown [194]. Though these treatments are known to lower the risk of spontaneous non-traumatic ICH, the validity of their use in TBI has been inadequately quantified. Retrospective study results [194,195,196] provided the earliest evidence for less operative interventions and decreased mortality in patients with blunt traumatic ICH associated with preinjury supplementation of DOACs rather than warfarin. Other common drugs such as selective serotonin reuptake inhibitors might also influence hemostasis [197]; however, their effect on TBI outcomes remains inadequately studied.

When comparing the blood product usage in anti-coagulated trauma patients with and without TEG-guided administration, blood product use was significantly lower when utilizing a TEG-guided approach; this is statistically independent from the mortality rates in both groups as they were extremely comparable [141]. In other words, a TEG-guided approach to reversing anticoagulation in TBI patients may improve the efficiency of blood product usage without harmfully affecting mortality [141]. Cartridge-based modified TEG^®^ with anti-Xa and direct thrombin inhibitor channels can provide DOAC levels within minutes [127]. Likewise, very recent ClotPro specific DOAC channels have been used to also determine levels of DOACs at the bedside in a comparatively short time. However, clinical research is still in its infancy for these tests. Although, it would be quite useful to know for patients with TBI, the patient’s hemostatic competence of those on DOACs [198]. The use of PCC to reverse anti-Vitamin K oral antagonists in emergencies is a well-defined practice [47]. Bleeding complications seen with vitamin-K antagonists can be reduced through the use of DOAC-specific reversal agents. Beginning in 2015, Idarucizumab became available as a target reversal agent for dabigatran, a thrombin inhibitor [196]. Additionally, the reversal agent for factor Xa-inhibitors, andexanet alfa, has been recently introduced [72]. As mentioned above, to date there are few studies evaluating the new reversal agents with VETs [198].

## 5. Conclusions

In this review, we provide a detailed description of the literature regarding the utility of VETs in the diagnosis and treatment of CTBI. The management of CTBI patients is complicated by the scarcity of clinical data regarding the underlying pathophysiology and standard treatment strategies for CTBI [23,48,199,200].

Analysis of the benefits of VET-guided management of TBI and CTBI is in its infancy, yet it is important to acknowledge that significant gaps in knowledge persist. Similar gaps existed regarding the value of VETs to guide CT and HAT during trauma and hemorrhagic resuscitation of trauma and non-trauma situations, with a gradual affirmation of the utility of VETs in these situations, which has taken decades [42,201,202,203,204,205,206]. The degree of lesions in brain tissue and consequential hemostatic impairment contributes to the heterogeneity and complexity of a TBI injury, thus making it difficult to compare patients diagnosed with TBI. The inconsistent results produced in some studies documented in this review can, in part, be explained by this heterogeneity of TBI and of the methodology of defining CTBI. Thus, future studies ought to consider the heterogeneity of TBI and CTBI patients in their analyses. The pathophysiologic evolution of CTBI changes rapidly; therefore, collection and analysis of blood sampling must be quick and efficient. The most optimal time to do this for TBI and CTBI patients is promptly at emergency room admission. Finally, hemostatic interventions guided by TEG-PM^®^, like administration of desmopressin and transfusion of platelet concentrate, to treat TBI-related platelet dysfunction and CTBI have not been subjected to rigorous analysis. The use of these strategies to treat patients with CTBI presents an interesting future in the history of VET-guided resuscitation for patients with CTBI.

## Figures and Tables

**Figure 1 jcm-10-05039-f001:**
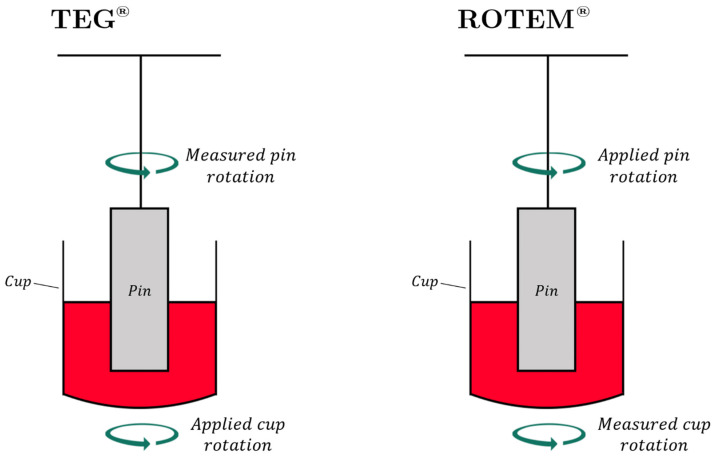
The TEG^®^ and ROTEM^®^ analyzers are each composed of a cup containing a whole blood sample, a pin suspended in the blood sample, a torsion wire, and a transducer. The cup is rotated at a speed of 4.45° per 10 s in TEG^®^. In ROTEM^®^, the pin is instead rotated, at the same speed of 4.45° per 10 s. In both assays, clotting of the whole blood gradually synchronizes the rotations of the cup and pin, which causes a change in torque on the torsion wire that is measured by the transducer. Various coagulation activators may be used depending on the assay. Intrinsic coagulation activators include kaolin or ellagic acid; extrinsic activation most commonly uses tissue factor [43,106,107].

**Figure 2 jcm-10-05039-f002:**
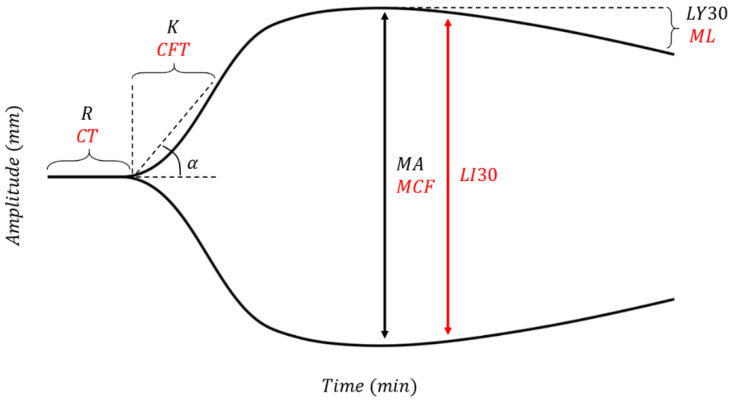
A typical graphical output of TEG^®^ (parameters shown in black) and ROTEM^®^ (parameters shown in red). R (Reaction Time)/CT (Clotting Time) denotes the time taken for blood to begin initiation of enzymatic clotting factor activation (marked by a movement of 2 mm along the *y*-axis). K/CFT (Clot Formation Time) denotes the time taken for movement of the pin by 20 mm along the *y*-axis. The α-angle is software-calculated using the slope of the secant line from the split point of the curve to K [111]. Clot kinetics are typically determined by K and α-angle, which together describe clot-strengthening rate and the cleavage of fibrinogen to fibrin by thrombin. Maximum Amplitude (MA) or Maximum Clot Firmness (MCF) denotes the peak of the curve and the point of greatest platelet-fibrin interaction [105,107]. Lysis at 30 min (LY30) is measured 30 min after MA as a percentage dissolution from MA peak. LI30 (Lysis Index at 30min) is measured as the amplitude 30 min after CT. Maximum Lysis (ML) is expressed as a percentage dissolution from MCF peak at the time of evaluation during the performance of the test and is roughly equivalent to the LY30 [43,106,107,108,112].

**Figure 3 jcm-10-05039-f003:**
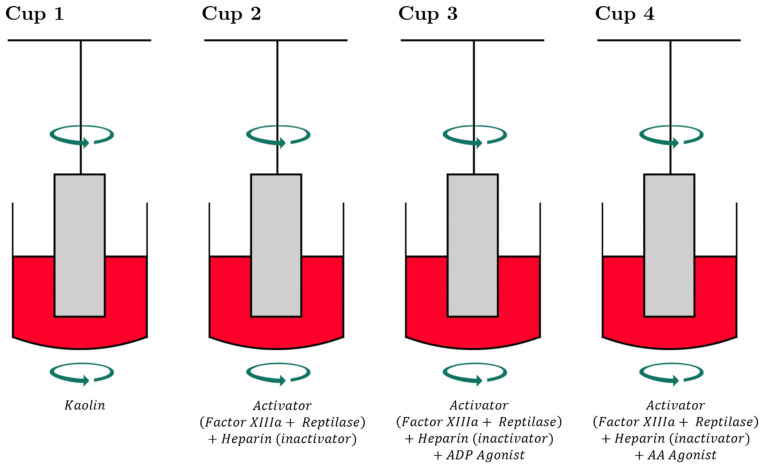
The analysis of platelet function requires four cups for Modified Thromboelastography with Platelet Mapping (TEG-PM^®^). In Cup 1 is a baseline kaolin TEG^®^ which describes control parameters. In Cups 2–4, heparin is added to neutralize thrombin which allows for isolation of platelet function in the presence of a pure fibrin clot. Reptilase and Factor XIIIa are added to Cups 2-4 to enhance fibrinogen and fibrin formation. Therefore, the additions of ADP in Cup 3 and AA in Cup 4 allow selective and respective activation of isolated ADP and AA receptors which then create an isolated pure fibrinogen/fibrin-platelet clot [109,121].

**Figure 4 jcm-10-05039-f004:**
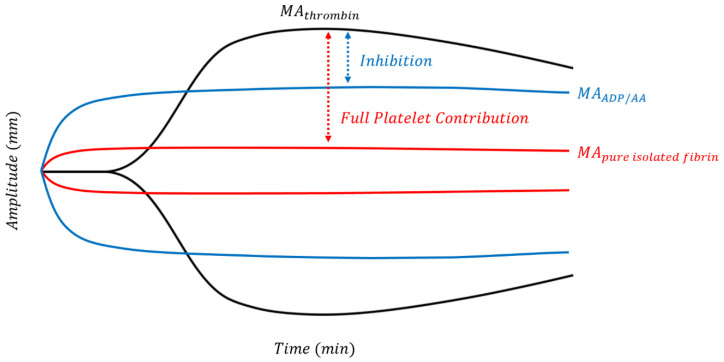
The TEG^®^ PlateletMapping assay is composed of four tests, starting with a standard kaolin TEG^®^ (black line; corresponds to Cup 1 of Figure 3) that depicts maximally activated platelets and full clot strength potential (MA_thrombin_). The clot strength of pure fibrin (red line; corresponds to Cup 2 in Figure 3) is isolated after blockage of all thrombin in the second test. Platelets are then activated in the third and fourth tests through either their ADP or AA receptors, and clot strength of platelets activated at either receptor (blue line; corresponds to Cups 3 & 4 in Figure 3) is evaluated by the proximity of the stimulated platelet as a percentage of MA_thrombin_ [128]. The platelet inhibition in response to the agonist is calculated from platelet aggregation: [(MA_ADP_ − MA_fibrin_)/(MA_thrombin_ − MA_fibrin_) × 100] and % inhibition = (100 − %aggregation) [109].

**Table 1 jcm-10-05039-t001:** Studies using modified VETs or VETs with adjunctive platelet function analyses in TBI. AA, arachidonic acid; ADP, adenosine diphosphate; AIS_head_, abbreviated injury severity score, head; DDAVP, desmopressin; GCS, Glasgow Coma Scale; INR, international normalized ratio; ISS, injury severity score; MEA, multiple electrode aggregometry; ROTEM, rotational thromboelastometry; TBI, traumatic brain injury; TEG-PM, thromboelastography with platelet mapping; TRAP, thrombin receptor activating peptides; VET, viscoelastic test.

Reference	Study Design (VET Used)	No. of Patients	Conclusions
Nekdulov et al., 2007 [84]	Prospective Observational (TEG-PM)	20 isolated TBI (GCS < 8, AIS-non-head ≤ 3)	TBI patients had 78% AA inhibition compared to 27% AA inhibition for healthy controls. The 8/20 TBI patients that bled had a significantly greater AA inhibition than nonbleeders.
Solomon et al., 2011 [65]	Retrospective Observational (ROTEM, MEA)	163 polytrauma	Mortality was correlated with low platelet aggregation by ADPtest, TRAPtest, and ROTEM platelet component contribution.
Wohlauer et al., 2012 [83]	Retrospective Observational (TEG-PM)	10 polytrauma TBI	Patients with TBI had a median ADP inhibition of 89.4% and median AA inhibition of 40.1% despite normal platelet counts and INR.
Davis et al., 2013 [1]	Retrospective Observational (TEG-PM)	50 isolated TBI (AIS_head_ ≥ 3, AIS-non-head < 2)	The median ADP inhibition was >91.7% for nonsurvivors vs. 48.2% for survivors; however, this difference was not statistically significant.
Castellino et al., 2014 [53]	Retrospective Observational (TEG-PM)	70 isolated TBI (AIS_head_ ≥ 3, AIS-non-head < 2)	The median ADP receptor inhibition of all TBI patients was 64.5% vs. 15.5% in controls. For GCS ≤ 8, the median ADP inhibition was 93.1% vs. 56.5% for those with GCS > 8. The median AA inhibition of all TBI patients was 25.6% vs. 2.2% in healthy controls.
Daley et al., 2017 [151]	Retrospective Observational (TEG-PM)	90 isolated and polytrauma TBI (AIS_head_ ≥ 3)	Patients with ADP inhibition on TEG-PM had a higher in-hospital mortality rate (8% vs. 32%). After controlling for age, gender, hypotension, GCS, ISS, and preinjury antiplatelet use, ADP inhibition > 60% demonstrated a significant odds ratio for mortality. AA inhibition > 60% was not found to be significant.
Furay et al., 2018 [61]	Retrospective Case-Control (TEG-PM)	35 isolated and polytrauma blunt TBI (AIS_head_ ≥ 3)	Patients who received TEG-PM guided goal-directed platelet transfusion for ADP inhibition > 60% had a significantly lower mortality compared to those who received no platelet transfusions (9% vs. 35%).
Guillotte et al., 2018 [64]	Retrospective Observational (TEG-PM)	153 TBI	ADP inhibition was greater in moderate/severe TBI compared to mild TBI. ADP inhibition was not found to be associated with mortality or intracerebral lesion expansion. There was no significant difference in the reduction of ADP inhibition with platelet transfusion compared to patients who did not receive platelet transfusion.
Kay et al., 2019 [144]	Retrospective Observational (TEG-PM)	119 isolated TBI (AIS_head_ ≥ 3, AIS-non-head < 2)	The median ADP inhibition was 18.4 points higher in severe TBI (AIS_head_ = 4 or 5) compared to moderate TBI (AIS_head_ = 3). Increased degree of ADP inhibition was also associated with increased odds of in-hospital mortality.
Furay et al., 2020 [62]	Retrospective Observational (TEG-PM)	57 isolated and polytrauma blunt TBI with ICH (AIS_head_ ≥ 3)	There was no difference in post-treatment ADP inhibition levels whether DDAVP alone or platelets alone were administered, guided by TEG-PM ADP inhibition > 60% as threshold for therapy. There was no significant difference in all-cause mortality between the two treatment groups.

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
