# Peer review of "Viscoelastic Testing and Coagulopathy of Traumatic Brain Injury"

_jcm, 2021, doi:10.3390/jcm10215039_

Round 1
Reviewer 1 Report
The authors of the manuscript focused on global hemostasis tests in traumatic brain injury. The authors have come up with an interesting review that is very important for medical research. Rotational thromboelastometry and thromboelastography is a holistic blood coagulation assay. Assessment of the immediate need for specific blood product transfusions in acutely bleeding patients is challenging. Clinical assessment and commonly used coagulation tests are inaccurate and time-consuming. Using TEG/ROTEM-guided blood transfusions in acutely bleeding trauma, surgical, and critically ill patients were associated with a tendency to fewer blood product transfusions in all populations. Some parts of the manuscript need to be corrected and supplemented in order for this manuscript to be published.
Page 2, lines 86-98. Global hemostatic assays provide information on the hemostatic efficacy based on the surrogate end point of maximum clot firmness and allow the evaluation of additional aspects such as the kinetics of clot formation, fibrin – platelet interactions, and the rate of fibrinolysis. These important facts should be mentioned and at the same time cite the manuscript for which it was published: ,,Semin Thromb Hemost. 2016 Jun;42(4):356-65. doi: 10.1055/s-0036-1571340“
Pages 8-9, lines 300-111. Overall, fibrinogen concentrate infusions are efficient and use should be primarily for bleeding. If unavailable, FFP or cryoprecipitate could be used in the second intention. Fibrinogen replacement could be guided by MCF values using rotational thromboelastometry. This statement was published in a manuscript which should be quoted: ,, Thromb Res. 2020 Apr;188:1-4. doi: 10.1016/j.thromres.2020.01.024.“
Page 9, lines 339-352: In this part, the authors should mention the important fact that fibrinogen plays a key role in hemostasis: fibrinogen conversion to fibrin, fibrin assembly, crosslinking, and platelet interactions. These important facts were published in a manuscript that should be cited: ,, J Thromb Thrombolysis. 2020 Jul;50(1):233-236. doi: 10.1007/s11239-019-01991-x.“
Figures and tables in the text are very clearly written.
I have to say that with these 203 references much of the references consists of manuscripts that have been published in the last 5 years
Author Response
Please see the pdf attachment. Thank you.

Reviewer 2 Report
The submitted manuscript by Bradbury et al. tried to give an overview about the usage and benefits of viscoelestic testing in traumatic brain injury. Overall I expected a better manuscript with 20! authors. Here are my major issues with the manuscript:
Major comments:
1/ Mostly, I take issues how the manuscript is written. A part of the manuscript cites the literature after every statement and in other parts literature was cited at the end of paragraphs. I would have expected that with 20 authors someone smooths the text to make it consistent. Please state the cited literature after each statement and not at the end of the paragraphs.
2/ Also, it is not clear why there are so many authors on this review.
3/ Line 34: What is hemo-coagulative?
4/ Line 48: What are the different definitions of coagulopathy the author referring to? To keep in mind, nomenclature refers to coagulopathy as inability/reduced blood clot formation (plasma based).
5/ Line 107 and following: Are the authors referring to platelet exhaustion? Please discuss. Also "coagulation" refers to the blood coagulation cascade and not platelets. The authors should use this term more carefully.
6/ Line 122: vWF mediates rather platelet activation and not coagulation. Please correct.
7/ Line 123: Please provide more information, how brain-born TF activates the "production" of uncleaved vWF. Are the authors implying that TF leads to synthesis of vWF or do the authors want to say that TF causes release of pre-existing vWF?
8/ Line 128: What do the authors mean with "unbound TF"?
9/ Line 129: See above: What is hypocoagulopathy?
10/ Line 134: What do the authors mean with "increased hemostatic dearangment"?
11/ Line 153: How is the TEG/ROTEM assay triggered/initiated?
12/ Line 177: Please add heparin to the figure 3 cup 2-4.
13/ Line 185: Please indicate cup 1-4 from figure 3 in figure 4.
14/ Line 210: In the impedance platelet assay. The electricity does not activate the platelets. The platelets get activated by the specific agonists and then adhere/stick to the electrodes which reduces the impedance. In addition, PAR1 (receptor) and TRAP (agonist) or do the authors refer with TRAP also to a PAR4 agonist? In general, the authors should mention at least one time the used agonist and the receptor. Later the authors can just refer to the used agonist in their assays.
15/ Line 262: If the authors refer to trial. Please add the trial identification number.
16/ Line 302: Please add reference for this statement.
17/ Line 283-284: Please add reference for this statement.
18/ Line 274 and Line 541: ACT is activated clotting time not long ACT. Please correct.
19/ Line 224: See above with regard to Hyper and Hypo-coagulable platelets.
20/ Line 309: What do the authors mean with "algorithms for guiding platelets for patients"?
21/ Line 316: Please add refences since the authors mention studies.
22/ Line 339: Please do not call fibrinogen coagulation factor I, if then call TF coagulation factor III.
23/ Line 347: Please add a statement why in late stage TBI the plasma fibrinogen levels increase.
24/ Line 357: TF initiates the blood coagulation and not platelets. Platelets contribute enhance the reaction by providing a negatively charged surface. Please correct.
25/ Line 366 : Again PAR1 is the receptor and TRAP the agonist. Pick one.
26/ Line 387: Please add reference for "... TBI have also been studied".
27/ Line 409: Please add refences for "... platelet transfusion with such injuries".
28/ Line 465: Please refer to Warfarin and DOACs as anticoagulants.
29/ Line 509: See above, thrombin is usually called thrombin and not factor IIa, earlier the authors called fibrinogen coagulation factor I and not factor I. The 20! authors should stay consistent within the text.
Author Response

(The authors gave the same response as above.)

Round 2
Reviewer 1 Report
The presented manuscript has been corrected in response to the suggestions. The authors have followed the recommendations of the reviewer. After the revision, the provided data and addition of the results became more clear. I would like to thank the authors for resubmitting the manuscript and explaining the obscure points from the previous version.